# SimSiam Naming Game: A Unified Approach for Representation Learning and Emergent Communication

## Abstract

Emergent communication, driven by generative models, enables agents to develop a shared language for describing their individual views of the same objects through interactions. Meanwhile, self-supervised learning (SSL), particularly SimSiam, uses discriminative representation learning to make representations of augmented views of the same data point closer in the representation space. Building on the prior work of VI-SimSiam, which incorporates a generative and Bayesian perspective into the SimSiam framework via variational inference (VI) interpretation, we propose SimSiam+VAE, a unified approach for both representation learning and emergent communication. SimSiam+VAE integrates a variational autoencoder (VAE) into the predictor of the SimSiam network to enhance representation learning and capture uncertainty. Experimental results show that SimSiam+VAE outperforms both SimSiam and VI-SimSiam. We further extend this model into a communication framework called the SimSiam Naming Game (SSNG), which applies the generative and Bayesian approach based on VI to develop internal representations and emergent language, while utilizing the discriminative process of SimSiam to facilitate mutual understanding between agents. In experiments with established models, despite the dynamic alternation of agent roles during interactions, SSNG demonstrates comparable performance to the referential game and slightly outperforms the Metropolis-Hastings naming game.

## 1 Introduction

Emergent communication (EmCom) studies how multiple agents, through interaction, can develop a shared language, known as a symbol emergence system (Cangelosi & Parisi, 2002; Taniguchi et al., 2016; 2019; Lazaridou & Baroni, 2020; Rita et al., 2024; Peters et al., 2024). Many studies in EmCom, based on Shannon-Weaver-like communication models (Shannon & Weaver, 1949), such as the Lewis signaling game (Lewis, 2008) or the referential game (Lazaridou et al., 2017), primarily focus on how agents can discriminate target objects or analyze the compositionality of the emergent signals (Havrylov & Titov, 2017; Denamganaï et al., 2023; Lipinski et al., 2024), often without considering internal representations. In contrast, collective predictive coding (CPC)-based EmCom (Taniguchi, 2024), such as the Metropolis-Hastings naming game (MHNG) (Hagiwara et al., 2019; Taniguchi et al., 2023b), views EmCom as a form of decentralized Bayesian inference. This approach focuses on both the representations learned within individual agents and the emergence of symbols at a societal level, referred to as social representation learning.

Representation learning, on the other hand, has been a fundamental aspect of machine learning (Bengio et al., 2013a; LeCun et al., 2015), particularly in tasks like image classification, where the objective is to extract meaningful features from raw data (Bishop, 2006). Within this domain, self-supervised learning (SSL) has attracted significant attention by enabling models to learn representations without relying on labeled data (Liu et al., 2021; Uelwer et al., 2023). One important approach in SSL is contrastive learning, which focuses on learning by comparing different augmented views of the same data point (Le-Khac et al., 2020). Notable models in this area, such as SimCLR (Chen et al., 2020), DINO (Caron et al., 2021), and SimSiam (Chen & He, 2021), have shown that this approach can align representations and improve feature extraction.

Both CPC-based EmCom and contrastive-based SSL follow a similar process. In CPC-based Em-Com, agents observe the same object from different viewpoints and iteratively develop a common language by aligning their internal representations through generative modeling (Taniguchi, 2024). In contrast, contrastive-based SSL models, particularly SimSiam, align augmented views of the same data point in the representation space through a discriminative process, relying only on positive pairs (Chen & He, 2021). Furthermore, recent research (Nakamura et al., 2023) has applied variational inference (VI) to SSL models, providing a generative interpretation of traditionally discriminative methods, such as SimSiam, and capturing uncertainty in learned representations.

Building on the VI-based interpretation of SSL models, we propose a unified approach that connects discriminative SSL-based representation learning with generative CPC-based EmCom. We introduce SimSiam+VAE, which integrates a Variational Autoencoder (VAE) (Kingma & Welling, 2013) into the predictor of the SimSiam network. This integration enhances latent representations with uncertainty by combining two processes: aligning positive pairs through contrastive comparison and refining representations via the VAE's encoding-decoding process, all without relying on negative samples.

We further extend SimSiam+VAE into a structured communication framework called the SimSiam Naming Game (SSNG), designed to facilitate EmCom between agents. In SSNG, each agent operates a separate SimSiam+VAE network, where the backbone and projector function as a perception module to transform observations into internal representations. The VAE predictor acts as a language coder, responsible for generating and decoding messages. Agents perceive different viewpoints of the same object and use a Bayesian approach to form internal representations and develop an emergent language. Through iterative exchanges, they interact similarly to the SimSiam+VAE, using its discriminative process to align their representations and achieve mutual understanding.

For evaluation, we conduct two experiments. First, we assess the performance of SimSiam+VAE in representation learning by measuring classification accuracy on the image datasets FashionMNIST and CIFAR-10. Second, we evaluate the SSNG's capability in emergent communication (EmCom) using the dSprites dataset, measuring the compositional generalization of the emergent language by applying TopSim (Brighton & Kirby, 2006) to unseen data (Chaabouni et al., 2020; Baroni, 2020)

Our contributions are summarized as follows:

- We formulate SimSiam+VAE, a unified model that bridges representation learning and EmCom through a generative and discriminative framework. By integrating a VAE into the SimSiam architecture, we enhance latent representation learning and uncertainty modeling, using only positive pairs.

- We introduce the SimSiam Naming Game (SSNG), a novel communication game grounded in the principles of CPC. SSNG utilized the combined generative-discriminative approach of SimSiam+VAE to iteratively align internal representations and develop a shared emergent language.

## 2 PRELIMINARIES

**Self-Supervised Learning (SSL) as Variational Inference (VI):** Recent work (Nakamura et al., 2023) suggests that SSL can be interpreted through the lens of VI, a probabilistic framework for learning latent variable models (Blei et al., 2017). In SSL, representations are typically learned by minimizing a contrastive loss between different augmented views of the same data, with the aim of bringing these views closer in their latent space representation. This process is analogous to VI, where augmented views are treated as "observations" that contribute to learning a shared latent variable. The augmentations in SSL function similarly to distinct modalities within a multimodal generative model in VI.

Denote $\mathbb{X} = x_A, x_B$, where $x_A$ and $x_B$ are two augmented views of the same data point. Fig. 1(a) illustrates the probabilistic graphical model (PGM), where the latent variable $z$ represents a shared representation of the augmented data. The objective of SSL, when viewed through VI, is to find parameters $\theta$ that maximize the likelihood of the observations given $z$. However, computing the true posterior $p_\theta(z|\mathbb{X})$ directly is intractable, leading to the use of a variational distribution $q_\phi(z|\mathbb{X})$ to

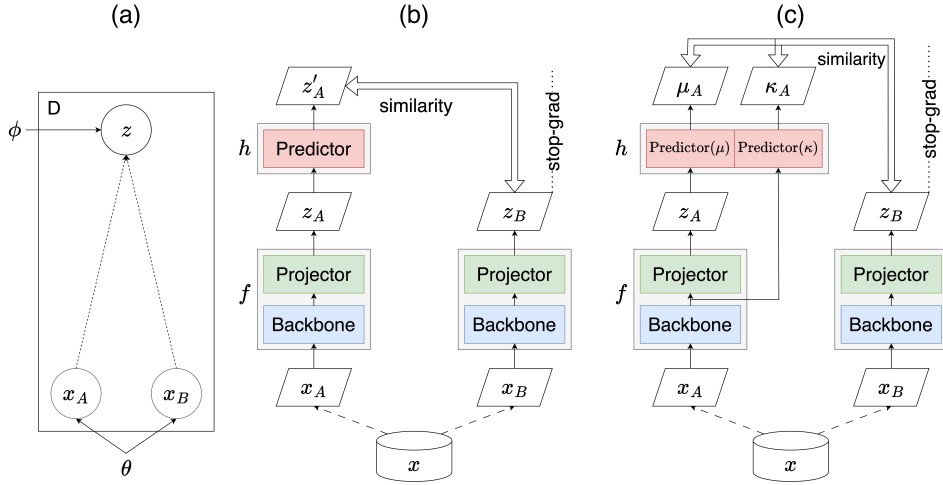

Figure 1: Illustrations of the SSL interpreted as a form of VI.
**(a)**: The PGM representation of the inference process in SSL. Observations $x_A$ and $x_B$ represent two augmented views (considered as multimodal observations) of the same data sample, derived from a dataset $D$. The arrows point from $x_A$ and $x_B$ to the latent variable $z$, indicating that the augmented views share a common latent representation $z$, which is inferred from these observations.
**(b)**: The SimSiam framework (Chen & He, 2021). Two augmented views, $x_A$ and $x_B$, are processed through a shared backbone and projector network $f$ to produce latent representations $z_A$ and $z_B$. A predictor network $h$ generates a transformed representation $z'_A$, which is compared to $z_B$ using a similarity measure. A stop-gradient operation is applied to $z_B$ to prevent gradient flow from $z'_A$, ensuring stable training and avoiding model collapse.
**(c)**: The proposed VI-SimSiam framework (Nakamura et al., 2023) extends SimSiam by modeling representation uncertainty. Latent representations $z_A$ and $z_B$ are produced similarly, but two predictors output the mean direction $\mu$ and concentration parameter $\kappa$ of the power spherical distribution, enabling both the representation and its uncertainty to be modeled.

approximate the posterior. This formulation leads to the objective function given by:

$$\mathbb{E}_{p(z|\mathbb{X})}[\log p_\theta(\mathbb{X}|z)] \geq \mathcal{J}_{\text{SSL}} := \mathbb{E}_{q_\phi(z|\mathbb{X})}[\log p_\theta(\mathbb{X}|z)] - D_{\text{KL}}[q_\phi(z|\mathbb{X})\|p(z)] \tag{1}$$

The SSL objective function is then decomposed as:

$$\mathcal{J}_{\text{SSL}} := \mathcal{J}_{\text{align}} + \mathcal{J}_{\text{uniform}} + \mathcal{J}_{\text{KL}} \tag{2}$$

where $\mathcal{J}_{\text{align}}$ encourages the alignment of representations from different views of the same data point, bringing them closer in the latent space. This aligns with the goal of SSL to learn invariant representations across augmented views. $\mathcal{J}_{\text{uniform}}$ promotes a well-distributed representation over the latent space to avoid collapse. Finally, $\mathcal{J}_{\text{KL}}$, introducing a Kullback-Leibler (KL) divergence, regularizes the approximate posterior distribution $q(z|\mathbb{X}, \phi)$ to be close to the prior $p(z)$.

The paper further demonstrates that specific SSL methods, such as SimSiam, SimCLR, and DINO, can be viewed under this VI framework by appropriately defining how they address alignment, uniformity, and regularization of latent variables. The inference process for these models operates as follows:

$$z \sim q_\phi(z|\mathbb{X}) = q_\phi(z|x_A, x_B) \qquad (z \text{ is inferred from both } x_A \text{ and } x_B) \tag{3}$$

## 3 SIMSIAM+VAE FOR REPRESENTATION LEARNING

### 3.1 MODEL DESCRIPTION

The proposed SimSiam+VAE model (Fig. 2) extends SimSiam by integrating a VAE into the predictor. The backbone and projector network, denoted as $f$, serves as a feature extractor, mapping

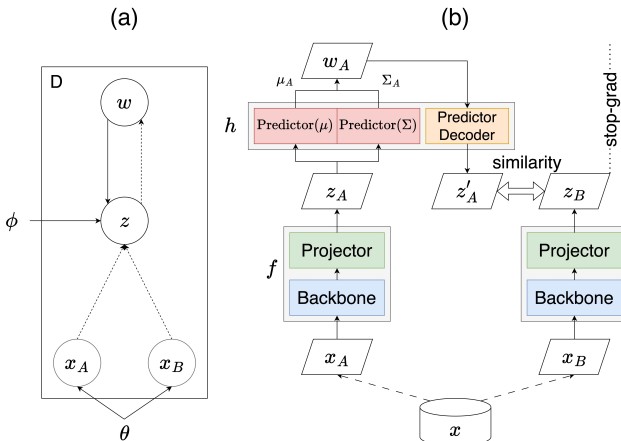

Figure 2: Illustrations of the SimSiam+VAE.

**(a)** The PGM representation of the generative and inference process in SimSiam+VAE. From the observations $x_A$ and $x_B$, the representation $z$ is inferred, which is subsequently used to infer latent variable $z$. Solid lines indicate the generative process (from $w$ to $z$), while dashed lines indicate the inference process (from $x_A$ and $x_B$ to $z$ and then to $w$).

**(b)** Architecture of the SimSiam+VAE framework. Two augmented views, $x_A$ and $x_B$, are processed through a shared backbone and projector network $f$ to produce representations $z_A$ and $z_B$. The predictor $h$ incorporates VAE components: the predictor encoder outputs the mean $\mu_A$ and covariance $\Sigma_A$ of the distribution over the latent variable $w_A$. The predictor decoder reconstructs the representation $z'_A$ from $w_A$. The similarity between $z'_A$ and $z_B$ is measured, and a stop-gradient operation is applied to $z_B$ to prevent collapse.

the augmented data $x_i$, for $i \in \{A, B\}$, to the latent representation $z_i$. The predictor, $h$, includes an encoder $h^{(\text{enc})}$ that maps $z_i$ to the parameters of a Gaussian distribution over a latent variable $w_i$, from which $w_i$ is sampled. The decoder $h^{(\text{dec})}$ then reconstructs $w_i$ to $z'_i$. The overall model, $g$, represents the composition of the backbone-projector network $f$ followed by the predictor $h$, such that $g = h \circ f$.

The inference process of the SimSiam+VAE operates as follows:

$$z \sim q_\phi(z|\mathbb{X}) = q_\phi(z|x_A, x_B) \qquad (z \text{ is inferred from both } x_A \text{ and } x_B) \qquad (4)$$

$$w \sim q_\phi(w|z) \qquad (w \text{ is inferred from } z) \qquad (5)$$

Similar to SimSiam, the proposed SimSiam+VAE model uses the stop-gradient mechanism to block gradients from being backpropagated through one of the branches. This mechanism treats the second latent representation as a constant, avoiding collapse to trivial solutions. Additionally, the VAE introduces a regularization term via the KL divergence, further preventing collapse through its encoding-decoding process. We conducted experiments to compare the model's performance with and without the stop-gradient mechanism, as discussed in Section 5.1.

## 3.2 OBJECTIVE FUNCTIONS

Fig. 2(a) shows the PGM of SimSiam+VAE, showing the inference process from augmented data $x_A$ and $x_B$ to representation $z$, and subsequently to the latent variable $w$. The objective of this model, under VI, is to find a parameter $\theta^*$ that maximizes the likelihood of the data. However, since the true posterior $p_\theta(\mathbb{X}|z, w)$ is intractable, we approximate it using the variational distribution $q_\phi(z, w|\mathbb{X})$. The resulting optimization problem is to maximize the objective function $\mathcal{L}_{\text{SSL}}$, which is defined as:

$$\theta^*, \phi^* = \arg\max_{\theta, \phi} \mathbb{E}_{q_\phi(z, w|\mathbb{X})} \left[ \log \frac{p_\theta(\mathbb{X}, z, w)}{q_\phi(z, w|\mathbb{X})} \right] \qquad (6)$$

This optimization leads to the objective function:

$$\mathcal{J}_{\text{SSL}} \approx \mathcal{J}_{\text{align}} + \mathcal{J}_{\text{recon}} + \mathcal{J}_{\text{uniform}} + \mathcal{J}_{\text{KL}} \qquad (7)$$

where

$$\mathcal{J}_{\text{align}} := \mathbb{E}_{q_\phi(z|\mathbb{X})}\left[\log p_\theta(z|\mathbb{X})\right] - \mathbb{E}_{q_\phi(z|\mathbb{X})}\left[\log q_\phi(z|\mathbb{X})\right] \tag{8}$$

$$\mathcal{J}_{\text{recon}} := \mathbb{E}_{q_\phi(z|\mathbb{X})}\left[\mathbb{E}_{q_\phi(w|z)}\left[\log p_\theta(z|w)\right]\right] \tag{9}$$

$$\mathcal{J}_{\text{uniform}} := \mathbb{E}_{q_\phi(z|\mathbb{X})}\left[-\log p_D(z)\right] \tag{10}$$

$$\mathcal{J}_{\text{KL}} := -\mathbb{E}_{q_\phi(z|\mathbb{X})}\left[D_{\text{KL}}\left(q_\phi(w|z,\mathbb{X})\|p(w)\right)\right] \tag{11}$$

$$p_D(z) := \mathbb{E}_{p_D(\mathbb{X})}[p_\theta(z|\mathbb{X})] \tag{12}$$

The alignment loss (Eq. 8) encourages the latent representations from different augmented views of the same data point to align in the representation space. The reconstruction loss (Eq. 9) encourages the VAE to accurately reconstruct the representation from the latent variable $w$. The uniform loss (Eq. 10) promotes a uniform distribution of representations in the representation space to avoid collapse. The KL-divergence term (Eq. 11) regularizes the distribution of the latent variable $w$, keeping it close to the prior. Lastly, (Eq. 12) defines the empirical distribution of the latent variables derived from the data.

In SimSiam+VAE, the prior $p(w)$ is a standard Gaussian distribution, while the prior $p(z)$ is uniform on the hypersphere $S^{d-1}$. The distribution $q_\phi(w|z)$ is modeled as a multivariate Gaussian distribution conditioned on $z$. Meanwhile, $p_\theta(z|w)$ is defined as a Dirac delta function, indicating a deterministic mapping from $w$ to $z$. The distribution $q_\phi(z|\mathbb{X})$ is modeled as a mixture of experts, where each expert corresponds to the contribution of an augmented view. The distribution $p_\theta(z|\mathbb{X})$ is represented as a product of experts, capturing the joint distribution across all augmented views:

$$p(w) \sim \mathcal{N}(0, I) \tag{13}$$

$$p(z) := \mathcal{U}(S^{d-1}) \tag{14}$$

$$q_\phi(w|z) := \mathcal{N}(w; \mu_w = h^{(\text{enc})}_{(\mu)}(z), \Sigma_w = h^{(\text{enc})}_{(\Sigma)}(z)) \tag{15}$$

$$p_\theta(z|w) := \delta(z - h^{(\text{dec})}(w)) \tag{16}$$

$$q_\phi(z|\mathbb{X}) := \frac{1}{M}\sum_{i=1}^{M}\delta(z - f_\phi(x_i)) \tag{17}$$

$$p_\theta(z|\mathbb{X}) := \eta_\theta \prod_{j=1}^{M}\text{vMF}(z; \mu_z = g_\theta(x_j), \kappa_z) \tag{18}$$

where

- $h^{(\text{enc})}_{(\mu)}$ and $h^{(\text{enc})}_{(\Sigma)}$ are the components of encoder network $h^{(\text{enc})}$ that generate the mean vector $\mu_w$ and covariance matrix $\Sigma_w$ of the Gaussian distribution from which $w$ is sampled.
- $h^{(\text{dec})}$ is are decoder of language coder $h$, providing a deterministic mapping from $w$ to $z$
- $\delta(z - f_\phi(x_i))$ is a Dirac delta function centered at $f_\phi(x_i)$.
- $\eta_\theta^{-1}$ is a normalization constant.
- $\text{vMF}(z; \mu_z, \kappa_z) := C_{\text{vMF}}(\kappa_z)\exp(\kappa_z \mu_z^\top z)$ is the von-Mises-Fisher distribution with mean direction $\mu_z$ and concentration parameter $\kappa_z \in \mathbb{R}^+$. The term $C_{\text{vMF}}(\kappa_z)$ is a normalization constant defined using the modified Bessel function. $\kappa_z$ is also constant.

The objective function of SimSiam+VAE is then given as:

$$\mathcal{J}_{\text{SSL}} \approx \sum_{i,j}\left(g_\theta(x_i)^\top f_\phi(x_j)\right) - \beta\sum_{i}D_{\text{KL}}\left(q_\phi(w|z,x_i)\|p(w)\right) \tag{19}$$

*Proof.* See Appendix B. □

In Eq. 19, the first term encourages the alignment of representations from different augmentations of the same input, similar to the reconstruction loss in a VAE. The second term is a regularization, ensuring that the latent variable $w$ remains close to the prior distribution. The hyperparameter $\beta$ controls the balance between the alignment and regularization terms, similar to the $\beta$-VAE introduced by Higgins et al. (2017). Pseudocode for the SimSiam+VAE model is provided in Appendix E.

# 4 SimSiam Naming Game for Emergent Communication

## 4.1 Model Description

The objective of SimSiam+VAE is to bring different views (augmentations) of the same data point closer in the representation space without relying on negative pairs. This aligns with the CPC-based EmCom, where two agents observe the same object from different viewpoints and develop shared representations without explicit labels. In this section, we extend SimSiam+VAE to facilitate EmCom between two agents, $A$ and $B$, through a communication game called the **SimSiam Naming Game (SSNG)**. Each agent $* \in \{A, B\}$ operates as a branch of the SimSiam+VAE, processing its observation $x_* \in \{x_A, x_B\}$, which is derived from a distinct viewpoint of the original object $x$.

Unlike the original SimSiam+VAE, which processes two augmentations of $x$ through a shared network to produce a single latent representation $z$ and a corresponding latent variable $w$, the SSNG introduces two separate latent representations, $z_A$ and $z_B$, one for each agent. Each branch of the network independently maps its observation $x_*$ to its internal representation $z_*$. These representations are then combined to form a shared message $w$, which acts as the emergent language for communication. The message $w$ enables the agents to align their internal representations, fostering mutual understanding. Through this structure, SSNG allows each agent to retain its unique perspective while contributing to a shared language. This approach aligns with Peirce's semiotics theory (Chandler, 2002), establishing a triadic relationship among the symbol (observation $x_*$), the interpretant (internal representation $z_*$), and the sign (message $w$) (Fig. 3).

Each agent $* \in \{A, B\}$ in this communication game has the two components: perception and language coder. The perception ($f_*$), consisting of the backbone and projector, transforms the observation $x_*$ into the internal representations $z_*$. The language coder ($h_*$) includes the predictor, which consists of an encoder ($h_*^{(\text{enc})}$) and a decoder ($h_*^{(\text{dec})}$). The encoder maps the internal representation $z_*$ to a shared message $w$ while the decoder receives and decodes the message into an internal representation $z_*'$.

The model components in the SSNG are identical to those in SimSiam+VAE. The key difference is that the latent variable $w$ now follows a categorical distribution over $K$, where $K$ represents the vocabulary or dictionary size. In the SSNG, the prior $p(w)$ is a uniform categorical distribution defined on the simplex $\Delta^{K-1}$ and $w$ is modeled as:

$$p(w) := \mathcal{U}(\Delta^{K-1}) \tag{20}$$

$$q_\phi(w|z) := \text{Cat}(w; \text{GS}(h_*^{(\text{enc})}(z))) \tag{21}$$

where $h_*^{(\text{enc})}(z)$ represents the logits produced from the internal representation $z$ via the encoder of language coder. These logits are converted into a categorical distribution, $\text{Cat}(w)$, using the Gumbel-Softmax (GS) distribution (Jang et al., 2017). The Straight-Through (ST) estimator is then applied to obtain one-hot vectors, enabling gradient-based training while maintaining discrete message representations (Bengio et al., 2013b).

## 4.2 Loss Function

In this communication game, agents A and B alternately take on the roles of speaker ($Sp$) and listener ($Li$), with possible role pairs $(Sp, Li) \in \{(A, B), (B, A)\}$. Given the listener ($Li$) and the message $w_{Sp}$ received from the speaker ($Sp$), the objective function of the listener is given by:

$$\mathcal{J}_{Li} \approx [h_{Li}^{(\text{dec})}(w_{Sp})]^\top f_{Li}(x_{Li}) - \beta D_{\text{KL}}(q_{Li}(w_{Li}|z_{Li}, x_{Li}) \| p(w_{Li})) \tag{22}$$

*Proof.* See Appendix C. □

This objective function is applied similarly for both agents A and B when either agent acts as the listener. In Eq. (22), the first term calculates the similarity loss between the decoded representation $z_{Sp}'$ (obtained from the received message $w_{Sp}$ through the decoder of listener's language coder $h_{Li}^{(\text{dec})}$) and the listener's internal representation $z_{Li}$ (generated by listener's perception $f_{Li}$). The second term serves as a regularization component that regularizes the listener's latent space $w_{Li}$.

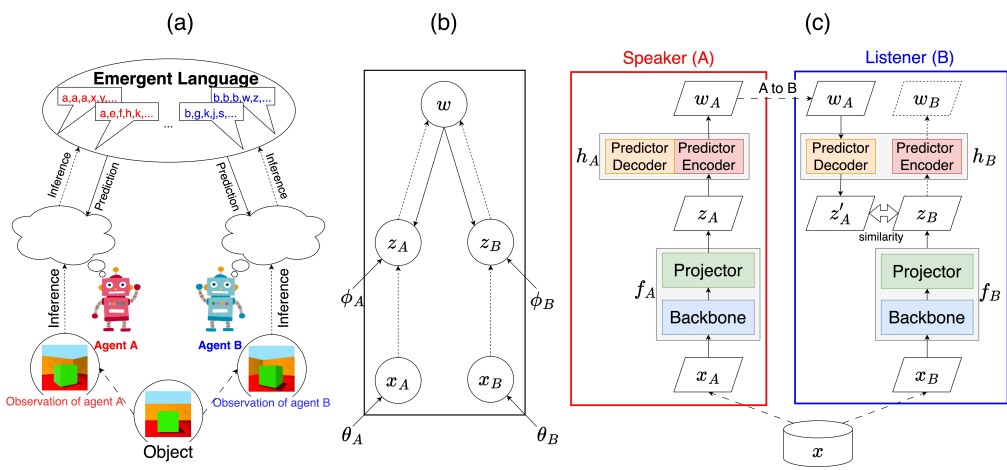

Figure 3: The EmCom between two agents, A and B, based on the SimSiam Naming Game.
**(a)** Two agents observe the same object from different perspectives. Each agent maps its observations to internal representations and uses them to infer and predict emergent language symbols, enabling them to communicate their perceptions and develop a shared emergent language.
**(b)** The PGM of SSNG: Denote agent $* \in \{A, B\}$. Solid lines represent the generative process, which starts from the shared latent variable $w$ to the representation $z_*$. Dashed lines represent the inference process, where each agent infers its representation $z_*$ from its observation $x_*$, and the shared message $w$ is inferred jointly from both agents' internal representations $z_A$ and $z_B$.
**(c)** The structure of agents: Both agents $*$ have the same model architecture with a backbone and projector $f_*$ and the predictor $h_*$ acts as the language coder, consisting of an encoder $h_*^{(\text{enc})}$ and a decoder $h_*^{(\text{dec})}$. In the example shown, agent A (depicted as the speaker) generates and transmits a message $w_A$ to agent B (as the listener), who processes it through a predictor decoder, producing an internal representation $z_A'$, which is then compared to $z_B$ to measure their similarity.

The inference process via the SSNG builds on the SimSiam+VAE with the parameters $\theta$ and $\phi$ spanning both agents: $\theta_A$, $\phi_A$ of agent A and $\theta_B$, $\phi_B$ of agent B. This process is detailed in Appendix D and operates as follows:

$$z_A \sim q_\phi(z_A|x_A) \qquad \text{(Agent A infers } z_A \text{ from } x_A) \qquad (23)$$

$$z_B \sim q_\phi(z_B|x_B) \qquad \text{(Agent B infers } z_B \text{ from } x_B) \qquad (24)$$

$$w \sim q_\phi(w|z_A, z_B) \qquad \text{(The shared latent variable } w \text{ is inferred from both } z_A \text{ and } z_B) \qquad (25)$$

### 4.3 THE SIMSIAM NAMING GAME (SSNG)

The SSNG facilitates communication and mutual understanding between agents through the following sequence of interactions:

    **i) Perception:** The speaker $(Sp)$ observes the input $x_{Sp}$ related to object $x$ to form an internal representation $z_{Sp}$ using its perception module $f_{Sp}$.

    **ii) Naming:** The speaker $(Sp)$ generates a message $w_{Sp}$ using the encoder $h_{Sp}^{(\text{enc})}$ of the language coder and sends this message to the listener $(Li)$.

    **iii) Communication:** Upon receiving the message $w_{Sp}$, the listener $(Li)$ decodes it into $z_{Sp}'$ using the decoder $h_{Li}^{(\text{dec})}$ of language coder.

    **iv) Learning:** The listener $(Li)$ calculates the loss using Eq. 22 by comparing $z_{Sp}'$ with its own $z_{Li}$ (generated by $f_{Li}$), then updates its model parameters to refine its understanding.

    **v) Turn-taking:** After the interaction, the roles of $Sp$ and $Li$ are swapped, and the process repeats from step **i)**.

This communication game, aligning with the principle of CPC, enables each agent to iteratively update its understanding based on the shared symbols through encoding, sharing, decoding, and

Table 1: Classification performance of different models on FashionMNIST and CIFAR-10.

| Model | FashionMNIST (Top-1) | CIFAR-10 (Top-2) |
|---|---|---|
| SimSiam | 82.95 | 59.24 |
| VI-SimSiam | 81.87 | 62.80 |
| SimSiam+VAE (no stop-grad) | 10.00 | 20.00 |
| SimSiam+VAE (ours) | **84.27** | **67.98** |

learning. A comparison among referential games (Lazaridou et al., 2017), Metropolis-Hastings naming game (Taniguchi et al., 2023b) and our SimSiam naming game is presented in Appendix A. The pseudocode for the SSNG is provided in Appendix F.

## 5 EXPERIMENTS AND DISCUSSIONS

This section presents two experiments to evaluate the proposed SimSiam+VAE model and SimSiam naming game. The source code for these experiments is available on GitHub[1].

### 5.1 EXPERIMENT 1: SIMSIAM+VAE IN REPRESENTATION LEARNING

**Datasets:** We use the FashionMNIST (Xiao et al., 2017) and CIFAR-10 (Krizhevsky, 2009) datasets.

**Model architecture:** A Convolutional Neural Network (CNN) backbone is used for FashionMNIST, while ResNet18 (He et al., 2016) is used for CIFAR-10. In both cases, the projector and predictor utilize a multi-layer perceptron (MLP) architecture. (More details in Appendix G)

**Linear evaluation:** All models are trained for 500 epochs. Then, a classifier is trained on the frozen representations obtained from the model using the training set labels and then evaluated on the test set. For FashionMNIST, Top-1 accuracy is reported, while for CIFAR-10, Top-2 accuracy is used.

**Comparison Models:** We compare our SimSiam+VAE model against SimSiam, VI-SimSiam.

**Results and Discussion:** (Table 1), the stop-gradient mechanism is essential for the proposed Sim-Siam+VAE framework. Without it, the model collapses to a trivial solution and fails to capture representation features. Our results show that SimSiam+VAE outperforms both SimSiam and VI-SimSiam, highlighting the advantage of integrating a VAE into the SimSiam. This integration enhances the model's ability to capture diverse features, leading to improved representation learning.

### 5.2 EXPERIMENT 2: SIMSIAM NAMING GAME IN EMERGENT COMMUNICATION

**Datasets:** We use the dSprites (Matthey et al., 2017) dataset, which consists of images of 2D shapes varying across different generative factors. Both agents are provided with the same data point but from different perspectives. The agents are then tested using a set of unseen data points to assess their generalization capabilities.

**Model Architecture:** The vocabulary size $|V|$ is 100, and the message length is 10. An MLP is used for the perception (backbone and projector). For the language coder's encoder, we use an LSTM to generate a discrete distribution over $V$ at each time step ($t$). This process is performed autoregressively, using the embedding of the discrete token sampled at the previous step ($t-1$) as input. The decoder consists of another LSTM that processes the sequence of embeddings recurrently, with the final hidden state as the output. (More details in Appendix H).

**Evaluation:** All models are trained for 1000 epochs. We use Topographical Similarity (TopSim) to evaluate how well the emergent language disentangles and aligns with the generative factors.

**Comparison Models:** We compare the emergent language from our SSNG with those of the referential (Xu et al., 2022) and Metropolis-Hastings naming games (Hoang et al., 2024a), all of which use the same LSTM-based models for generating and decoding messages.

---

[1]...

Table 2: TopSim of different communication games on the dSprites. The referential game produces a single TopSim value, while the other games produce separate values for each agent (A and B).

| Model | TopSim (A) | TopSim (B) |
|---|---|---|
| Referential Game | 0.22 | |
| Metropolis-Hastings Naming Game | 0.19 | 0.18 |
| SimSiam Naming Game (ours) | 0.22 | 0.18 |

**Results and Discussion:** (Table 2) Compared to the referential game, where agents are fixed as either message generators or interpreters, SSNG demonstrates comparable performance. However, compared to MHNG, where agents can both create and interpret messages, SSNG achieves slightly better results. These suggest that SSNG is a potential alternative approach for facilitating EmCom.

## 6  RELATED WORK

**Emergent Communication (EmCom)** examines how agents develop a shared language through interactions, drawing inspiration from cognitive science theories (Wagner et al., 2003; Steels, 2015). Research in multi-agent reinforcement learning (Foerster et al., 2016) demonstrated how agents could develop communication to optimize collective rewards. Comprehensive surveys of this field include (Galke et al., 2022; Brandizzi, 2023; Boldt & Mortensen, 2024). Recent studies have focused on CPC-based, which emphasizes joint attention in human communication (Okumura et al., 2023). The MHNG (Taniguchi et al., 2023b) utilizes decentralized Bayesian inference to achieve a consensus on shared symbols, aligning with predictive coding and world model (Hohwy, 2013; Friston et al., 2021; Taniguchi et al., 2023a). The MHNG has been applied in multimodal datasets using methods like Inter-MDM (Hagiwara et al., 2022) and Inter-GMM+MVAE (Hoang et al., 2024b). Moreover, MHNG has been extended to recursive multi-agent communication systems (Inukai et al., 2023) and integrated into multi-agent reinforcement learning (Ebara et al., 2023).

**Representation learning** is essential in machine learning tasks like image classification, allowing models to extract features from raw data (Goodfellow et al., 2016). SSL has become a popular method for learning representations without labels (Jing & Tian, 2020). A key SSL approach is contrastive learning, which aligns representations by comparing different augmented views of the same data point (Cole et al., 2022). MoCo (He et al., 2020) introduces a momentum encoder to maintain a queue of negative samples, while BYOL (Grill et al., 2020) eliminates the need for negative pairs, using a stop-gradient mechanism to avoid collapse. Recent research has combined VAE and contrastive learning to improve representation learning. CR-VAE adds contrastive regularization to the VAE objective (Lygerakis & Rueckert, 2023), while ContrastVAE employs a two-view approach with ContrastELBO for sequential recommendations (Wang et al., 2022). Noise contrastive estimation is used in (Aneja et al., 2021) to reweight the prior distribution. Contrastive VAEs (cVAE) focus on isolating salient features in datasets to refine latent space representation (Abid & Zou, 2019).

## 7  CONCLUSIONS

This research introduces the SimSiam Naming Game (SSNG) and SimSiam+VAE, a unified model that bridges discriminative contrastive SSL-based representation learning with generative CPC-based EmCom through the perspective of VI. Although originating from distinct domains, both SSL and EmCom share the goal of aligning representations—either by learning invariant representations from augmented data views in SSL or by developing a shared language between agents observing the same object from different perspectives. By bridging these objectives, our model demonstrates applicability to both representation learning and EmCom.

Our experiments show that SimSiam+VAE outperforms both SimSiam and VI-SimSiam in representation learning without requiring negative pairs. In EmCom, SSNG leverages the discriminative properties of SimSiam and the generative Bayesian perspective of the VI interpretation to align agents' internal representations, fostering mutual understanding and enabling the development of an emergent language. This work, therefore, provides an alternative communication framework for EmCom systems.

ACKNOWLEDGMENTS

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

## A    COMPARISON AMONG REFERENTIAL GAME, METROPOLIS-HASTINGS NAMING GAME AND SIMSIAM NAMING GAME

| Aspect | Referential Game | Metropolis-Hastings (MH) Naming Game | SimSiam Naming Game (SSNG) |
|---|---|---|---|
| **Objective** | Develop emergent language (EmLang) to refer to shared objects or concepts, focusing on communication accuracy. | Develop EmLang through probabilistic updates, optimizing mutual understanding using MH algorithm. | Develop EmLang through self-supervised learning (SSL), focusing on similarity between representations of agents. |
| **Communication method** | Speaker sends a message to refer to a target object among distractors. | Agents exchange messages and update beliefs through acceptance rate based on MH algorithm. | Agents exchange messages to align and convey information based on representation similarity. |
| **Learning Mechanism** | Grounded in shared perception, where agents learn communication through feedback based on correct or incorrect reference selection. | Probabilistic updates of beliefs and message proposals using MH algorithm, incorporating joint attention. | Contrastive learning SSL via variational inference to align representations of agents, incorporating joint attention. |
| **Agent Roles** | A fixed speaker and a listener with distinct roles (describing and selecting objects). | Both agents are capable of proposing and evaluating messages iteratively to align their beliefs. | Both agents are capable of proposing and evaluating messages iteratively to align their latent representations. |
| **Observations** | Both agents refer to a single viewpoint of each object in the context. | Agents have different viewpoints or observations of the same object. | Agents have different viewpoints of observations of the same object. |
| **Representation Space** | Not a primary focus. | Continuous internal representation space, updated probabilistically through message exchanges. | Continuous internal representation space, aligned through maximizing similarity between different viewpoints. |
| **Information Exchange** | Messages are shared to refer to specific target objects. | Messages are exchanged and evaluated based on the MH acceptance rate. | Messages are exchanged and evaluated based on an SSL objective function. |
| **Interaction Mode** | One-way interaction: speaker sends a message, and listener interprets it to select the target object. | Iterative, bidirectional interaction: both agents propose and receive messages. | Iterative, bidirectional interaction: both agents propose and receive messages. |

Table 3: Comparison between referential game (Lazaridou et al., 2017), Metropolis-Hastings naming game (MHNG) (Taniguchi et al., 2023b), and SimSiam naming game (SSNG).

# B  SimSiam+VAE - Objective function

The objective function of SimSiam+VAE is derived as follows:

$$\mathcal{L}_{\text{SSL}} := \mathbb{E}_{q_\phi(z,w|\mathbb{X})} \left[ \log \frac{p_\theta(\mathbb{X}, z, w)}{q_\phi(z, w|\mathbb{X})} \right] \tag{26}$$

$$:= \mathbb{E}_{q_\phi(z,w|\mathbb{X})} \left[ \log \frac{p_\theta(\mathbb{X} \mid z)p_\theta(z \mid w)p(w)}{q_\phi(w|z, \mathbb{X})q_\phi(z|\mathbb{X})} \right] \tag{27}$$

$$:= \mathbb{E}_{q_\phi(z,w|\mathbb{X})} \left[ \log p_\theta(\mathbb{X}|z) + \log p_\theta(z|w) + \log p(w) - \log q_\phi(w|z, \mathbb{X}) - \log q_\phi(z|\mathbb{X}) \right] \tag{28}$$

$$:= \mathbb{E}_{q_\phi(z,w|\mathbb{X})} \left[ \log p_\theta(\mathbb{X}|z) \right] - \mathbb{E}_{q_\phi(z,w|\mathbb{X})} \left[ \log q_\phi(z|\mathbb{X}) \right] +$$
$$+ \mathbb{E}_{q_\phi(z,w|\mathbb{X})} \left[ \log p_\theta(z|w) \right] + \mathbb{E}_{q_\phi(z,w|\mathbb{X})} \left[ \log p(w) - \log q_\phi(w|z, \mathbb{X}) \right] \tag{29}$$

Since $p_\theta(\mathbb{X})$ is intractable, we approximate it with empirical data distribution $p_D(\mathbb{X})$. Using Bayes' theorem:

$$p_\theta(\mathbb{X}|z) = \frac{p_\theta(z|\mathbb{X})p_\theta(\mathbb{X})}{\mathbb{E}_{p_\theta(\mathbb{X})}[p_\theta(z|\mathbb{X})]} \approx \frac{p_\theta(z|\mathbb{X})p_D(\mathbb{X})}{\mathbb{E}_{p_D(\mathbb{X})}[p_\theta(z|\mathbb{X})]} \tag{30}$$

then

$$\mathbb{E}_{q_\phi(z,w|\mathbb{X})} \left[ \log p_\theta(\mathbb{X}|z) \right] \tag{31}$$

$$\approx \mathbb{E}_{q_\phi(z,w|\mathbb{X})} \left[ \log \frac{p_\theta(z|\mathbb{X})p_D(\mathbb{X})}{\mathbb{E}_{p_D(\mathbb{X})}[p_\theta(z|\mathbb{X})]} \right] \tag{32}$$

$$\approx \mathbb{E}_{q_\phi(z,w|\mathbb{X})} \left[ \log p_\theta(z|\mathbb{X}) + \log p_D(\mathbb{X}) - \log \mathbb{E}_{p_D(\mathbb{X})}[p_\theta(z|\mathbb{X})] \right] \tag{33}$$

$$\approx \mathbb{E}_{q_\phi(z,w|\mathbb{X})} \left[ \log p_\theta(z|\mathbb{X}) \right] - \mathbb{E}_{q_\phi(z,w|\mathbb{X})} \left[ \log \mathbb{E}_{p_D(\mathbb{X})}[p_\theta(z|\mathbb{X})] \right] + \log p_D(\mathbb{X}) \tag{34}$$

Besides,

$$\mathbb{E}_{q_\phi(z,w|\mathbb{X})} \left[ \log p(w) - \log q_\phi(w|z, \mathbb{X}) \right] \tag{35}$$

$$= \mathbb{E}_{q_\phi(z|\mathbb{X})} \left[ \mathbb{E}_{q_\phi(w|z,\mathbb{X})} \left[ \log p(w) - \log q_\phi(w|z, \mathbb{X}) \right] \right] \tag{36}$$

$$= -\mathbb{E}_{q_\phi(z|\mathbb{X})} \left[ D_{\text{KL}} \left( q_\phi(w|z, \mathbb{X}) \parallel p(w) \right) \right] \tag{37}$$

Substituting Eqs. (34) and (37) to Eq. (29), the objective function is:

$$\mathcal{J}_{\text{SSL}} \approx \mathcal{J}_{\text{align}} + \mathcal{J}_{\text{recon}} + \mathcal{J}_{\text{uniform}} + \mathcal{J}_{\text{KL}} + \log p_D(X) \tag{38}$$

$$\approx \mathcal{J}_{\text{align}} + \mathcal{J}_{\text{recon}} + \mathcal{J}_{\text{uniform}} + \mathcal{J}_{\text{KL}} \tag{39}$$

where

$$\mathcal{J}_{\text{align}} := \mathbb{E}_{q_\phi(z|\mathbb{X})} \left[ \log p_\theta(z|\mathbb{X}) \right] - \mathbb{E}_{q_\phi(z|\mathbb{X})} \left[ \log q_\phi(z|\mathbb{X}) \right] \tag{40}$$

$$\mathcal{J}_{\text{recon}} := \mathbb{E}_{q_\phi(z,w|\mathbb{X})} \left[ \log p_\theta(z|w) \right] \tag{41}$$

$$\mathcal{J}_{\text{uniform}} := \mathbb{E}_{q_\phi(z|\mathbb{X})} \left[ -\log p_D(z) \right] \tag{42}$$

$$\mathcal{J}_{\text{KL}} := -\mathbb{E}_{q_\phi(z|\mathbb{X})} \left[ D_{\text{KL}} \left( q_\phi(w|z, \mathbb{X}) \| p(w) \right) \right] \tag{43}$$

$$p_D(z) := \mathbb{E}_{p_D(\mathbb{X})}[p_\theta(z|\mathbb{X})] \tag{44}$$

In SimSiam+VAE, we define $p(w)$, $p(z)$, $q_\phi(w|z)$, $p_\theta(z|w)$, $q_\phi(z|\mathbb{X})$, and $p_\theta(z|\mathbb{X})$ as mentioned in Eqs. (13), (14), (15), (16), (17), (18), respectively.

ALIGNMENT LOSS

$$\mathcal{J}_{\text{align}} := \mathbb{E}_{q_\phi(z|\mathbb{X})}\left[\log p_\theta(z|\mathbb{X}) - \log q_\phi(z|\mathbb{X})\right] \tag{45}$$

$$:= \frac{1}{M}\sum_{j=1}^{M}\left[\log p_\theta(f_\phi(x_j)|\mathbb{X}) - \log q_\phi(f_\phi(x_j)|\mathbb{X})\right] \tag{46}$$

$$:= \frac{1}{M}\sum_{j=1}^{M}\left[\log\left(\eta_\theta\prod_{i=1}^{M}\text{vMF}(f_\phi(x_j); \mu_z = g_\theta(x_i), \kappa_z)\right) - \log\frac{1}{M}\right] \tag{47}$$

$$:= \frac{1}{M}\sum_{j=1}^{M}\left[\log\eta_\theta + \log M + \sum_{i=1}^{M}\log\text{vMF}(f_\phi(x_j); \mu_z = g_\theta(x_i), \kappa_z)\right] \tag{48}$$

$$\mathcal{J}_{\text{align}} \approx \sum_{i,j}\left(g_\theta(x_i)^\top f_\phi(x_j)\right) \tag{49}$$

RECONSTRUCTION LOSS

$$\mathcal{J}_{\text{recon}} := \mathbb{E}_{q_\phi(z|\mathbb{X})}\left[\mathbb{E}_{q_\phi(w|z)}\left[\log p_\theta(z|w)\right]\right] \tag{50}$$

The inner term $\mathbb{E}_{q_\phi(w|z)}\left[\log p_\theta(z|w)\right]$ represents the reconstruction loss in the VAE component. In representation learning, this loss can be approximated by:

$$\mathbb{E}_{q_\phi(w|z)}\left[\log p_\theta(z|w)\right] \approx (z')^\top z = g_\theta(x)^\top f_\phi(x) \tag{51}$$

where $z'$ denotes the reconstructed representation obtained from the latent variable $w$. This approximation captures the alignment between the original and reconstructed representations in the representation space. Thus,

$$\mathcal{J}_{\text{recon}} \approx \sum_{i}\left(g_\theta(x_i)^\top f_\phi(x_i)\right) \tag{52}$$

The reconstruction loss $\mathcal{J}_{\text{recon}}$ measures the alignment between the reconstructed representation $g_\theta(x_i)$ and the original one $f_\phi(x_i)$. This alignment is already captured by the $\mathcal{J}_{\text{align}}$. Hence, $\mathcal{J}_{\text{recon}}$ is omitted from the total loss.

UNIFORM LOSS

The role of $\mathcal{J}_{\text{uniform}}$ is to ensure that the marginal distribution $p_D(z)$ is uniform over the hypersphere, i.e., $p_D(z) = \mathcal{U}(S^{d-1})$. However, the predictor $h$, defined as a DirectPred (Tian et al., 2021), ensures that the latent representations $z$ are uniformly spread over the hypersphere. It achieves this by making the distribution of $z$ approximately isotropic, with each dimension being independent and having equal variance. Consequently, $h$ implicitly maximizes $\mathcal{J}_{\text{uniform}}$ (Nakamura et al., 2023).

Since the predictor already encourages a uniform distribution of the representations, explicitly including $\mathcal{J}_{\text{uniform}}$ in the total loss is redundant. Therefore, it can be omitted without losing the intended effect on the representation distribution.

KL DIVERGENCE

Since each representation $z$ is derived from the same network with a stop-gradient operation, the KL divergence can be simplified as:

$$\mathcal{J}_{\text{KL}} \approx -\sum_{i} D_{\text{KL}}\left(q_\phi(w|z, x_i)\|p(w)\right) \tag{53}$$

TOTAL LOSS

Combining these components, the objective function of SimSiam+VAE is given by:

$$\mathcal{J}_{\text{SSL}} \approx \sum_{i,j}\left(g_\theta(x_i)^\top f_\phi(x_j)\right) - \beta\sum_{i} D_{\text{KL}}\left(q_\phi(w|z, x_i)\|p(w)\right) \tag{54}$$

## C    SimSiam Naming Game - Objective function

In the SimSiam naming game with two agents, A and B, the total loss function $\mathcal{J}_{\text{SSL}}$, derived from the objective function of SimSiam+VAE, is adapted to account for each agent's individual observations and representations. Unlike the original SimSiam+VAE, the SSNG separates $z$ into two latent representations, $z_A$ and $z_B$, one for each agent. Each agent $*$ receives a unique observation $x_*$, which is encoded into a representation $z_*$, and subsequently mapped to a shared latent variable $w$. The total loss is reformulated as:

$$\mathcal{J}_{\text{SSNG}} \approx \sum_{i,j} \left( g_\theta(x_i)^\top f_\phi(x_j) \right) - \beta \sum_i D_{\text{KL}} \left( q_\phi(w|z_A, z_B, x_i) \| p(w) \right) \tag{55}$$

This loss consists of the optimization process for both agent A and B. Therefore, the total loss can be decomposed into contributions for each agent:

$$\mathcal{J}_{\text{SSNG}} = \mathcal{J}_A + \mathcal{J}_B \tag{56}$$

where $\mathcal{J}_A$ and $\mathcal{J}_B$ represent the loss functions for agent A and agent B, respectively:

$$\mathcal{J}_A \approx g_B(x_B)^\top f_A(x_A) - \beta D_{\text{KL}} \left( q_A(w|z_A, z_B, x_A) \| p(w) \right) \tag{57}$$

$$\mathcal{J}_B \approx g_A(x_A)^\top f_B(x_B) - \beta D_{\text{KL}} \left( q_B(w|z_A, z_B, x_B) \| p(w) \right) \tag{58}$$

In the objective function of SimSiam+VAE, the parameters $\theta$ and $\phi$ are shared across all observations. When this objective is split into agent-specific losses, these parameters become agent-specific versions: $\theta_A$, $\phi_A$ for agent A and $\theta_B$, $\phi_B$ for agent B. For simplicity, we denote the functions with these parameters as $f_A$, $f_B$, etc., where the subscript "A" or "B" indicates the respective agent.

In this communication game, agents A and B alternately take on the roles of speaker $(Sp)$ and listener $(Li)$, with possible role pairs $(Sp, Li) \in \{(A, B), (B, A)\}$. Given the listener $(Li)$ and the message $w_{Sp}$ received from the speaker $(Sp)$, the objective function of the listener is given by:

$$\mathcal{J}_{Li} \approx g_{Sp}(x_{Sp})^\top f_{Li}(x_{Li}) - \beta D_{\text{KL}} \left( q_{Li}(w_{Li}|z_{Li}, z_{Sp}, x_{Li}) \| p(w_{Li}) \right) \tag{59}$$

In EmCom, agents are unable to observe each other's internal concepts, much like humans cannot directly access one another's thoughts. Therefore, the listener cannot access the speaker's function $g_{Sp}$. Instead, the listener interprets the message received from the speaker using its own decoder. To do this, we start from each agent's function $g_*$ which is composed as follows:

$$g_* = h_*^{(\text{dec})} \circ h_*^{(\text{enc})} \circ f_* \tag{60}$$

where:

- $f_*$ is the perception, consisting of a backbone and projector, processing the observation $x_*$ to obtain the internal representation $z_*$.

- $h_*^{(\text{enc})}$ is the encoder of the language coder $h_*$, mapping the representation $z_*$ to the latent variable $w_*$.

- $h_*^{(\text{dec})}$ is the decoder of the language coder $h_*$, reconstructing a representation $z_*'$ from the received message $w_{Sp}$.

As described in Section 4.3, the SSNG is follows these steps:

- The speaker generates a message $w_{Sp}$ from its observation $x_{Sp}$

$$w_{Sp} = h_{Sp}^{(\text{enc})}(f_{Sp}(x_{Sp})) \tag{61}$$

- The message $w_{Sp}$ is then transmitted to the listener, who decodes it to produce a reconstructed representation $z_{Sp}'$:

$$z_{Sp}' = h_{Li}^{(\text{dec})}(w_{Sp}) \tag{62}$$

Since the listener cannot access the speaker's component $g_{Sp}$, it uses the reconstructed representation $z'_{Sp}$ to interpret the speaker's intent. Thus, the function $g_{Li}$, which reflects the listener's interpretation, is composed as:

$$g_{Li} = h_{Li}^{(\text{dec})} \circ h_{Sp}^{(\text{enc})} \circ f_{Sp} \tag{63}$$

Besides, since the listener cannot access the speaker's internal representation $z_{Sp}$, the $D_{\text{KL}}$ will be calculated based on its own $z_{Li}$. As a result, the loss function for the listener is reformulated as:

$$\mathcal{J}_{Li} \approx [h_{Li}^{(\text{dec})}(w_{Sp})]^{\top} f_{Li}(x_{Li}) - \beta D_{\text{KL}}\left(q_{Li}(w_{Li}|z_{Li}, x_{Li}) \| p(w_{Li})\right) \tag{64}$$

By this formulation, the listener's loss emphasizes how well it can decode the speaker's shared message $w_{Sp}$ using its own representations, as well as regularizing its own latent space via the KL divergence. This captures the partial observability and the need for the listener to independently infer and interpret the shared emergent language.

## D  INFERENCE VIA SIMSIAM NAMING GAME

The goal of both SimSiam+VAE and the SSNG is to align the representations of different viewpoints of the same object. This alignment process ensures that observations of the same object from different perspectives are represented closely in the latent space. The training process, which minimizes an alignment loss and a reconstruction loss, gradually reduces the dissimilarity between the internal representations of both agents.

To achieve this, the objective function encourages the representations $z_A$ from agent A's observation $x_A$ and $z_B$ from agent B's observation $x_B$ to become more similar. As the alignment improves, we achieve the approximation:

$$p(w \mid z_A, z_B) \approx p(w \mid z_A) \approx p(w \mid z_B) \tag{65}$$

Therefore, through the SSNG, the model can learn a shared latent variable $w$ that captures the mutual understanding between the two agents. This shared understanding is derived from the aligned representations $z_A$ and $z_B$, which reflect different views of the same underlying object.

## E  PSEUDOCODES OF SIMSIAM+VAE

**Algorithm 1** Pseudocode of SimSiam+VAE, PyTorch-like

```
# projector and backbone f()
# predictor h with h_enc() and h_dec()

for x in loader: # load a minibatch x
    xA, xB = augmented(x), augmented(x) # augmentation
    zA, zB = f(xA), f(xB) # backbone + projector
    wA, muA, logvarA = h_enc(zA) # predictor encoder of A
    wB, muB, logvarB = h_enc(zB) # predictor encoder of B
    zA_recon = h_dec(wA) # predictor decoder of A
    zB_recon = h_dec(wB) # predictor decoder of B

    zA, zB = zA.detach(), zB.detach() # Stop-gradient
    loss_align = D(zA_recon, zB) + D(zB_recon, zA)
    loss_KL = KL(muA, logvarA) + KL(muB, logvarB)
    loss = loss_align + loss_KL # total loss
    loss.backward() # back-propagate
    update(f, h) # update parameters

def D(x, y): # negative cosine similarity
    x = normalize(x, dim=1)
    y = normalize(y, dim=1)
    return -(x * y).sum(dim=1).mean()
```

# F PSEUDOCODES OF SIMSIAM NAMING GAME

**Algorithm 2** Pseudocode of SimSiam naming game, PyTorch-like

```
1  # get observation of object x with input_A() and input_B()
2  # perception f_Sp() of Speaker and f_Li() of Listener
3  # predictor of Speaker h_Sp with h_Sp_enc() and h_Sp_dec()
4  # predictor of Listener h_Li with h_Li_enc() and h_Li_dec()
5
6  for x in loader: # load a minibatch x
7      x_A, x_B = input_A(x), input_B(x) # observations of x
8      SSNG(Sp = B, Li = A) # SSNG: B as speaker and A as listener
9      SSNG(Sp = A, Li = B) # SSNG: A as speaker and B as listener
10
11 def SSNG(Sp, Li): # SimSiam naming game
12     z_Sp, z_Li = f_Sp(x_Sp), f_Li(x_Li) # perception
13     w_Sp, _ = h_Sp_enc(z_Sp) # speaker creates message
14     w_Li, logits_Li = h_Li_enc(z_Li) # listener creates message
15     z1_Sp = h_Li_dec(w_Sp) # listener decodes received message
16     loss = D(z_Li, z1_Sp) + KL(logits_Li) # Total loss of listener
17     loss.backward() # listener back-propagates
18     update(f_Li, h_Li) # listener updates parameters
```

# G EXPERIMENT 1 - SIMSIAM+VAE IN REPRESENTATION LEARNING

DATASETS:

- **FashionMNIST** (Xiao et al., 2017) contains 70,000 grayscale images, each of size 28x28, representing 10 classes of objects with 60,000 training and 10,000 testing images.
- **CIFAR-10** (Krizhevsky, 2009) is a collection of 60,000 color images, each of size 32x32 and belonging to one of 10 different classes with 50,000 training and 10,000 testing images.

MODEL ARCHITECTURE:

- Backbone network:
  - FashionMNIST Backbone: A custom CNN with two convolutional layers: the first outputs 16 channels (kernel size 4, stride 2, padding 1), and the second doubles the channels. A fully connected layer maps the features to 512 dimensions.
  - CIFAR-10 Backbone: ResNet18 in its original form. Additionally, an alternative CNN backbone is implemented with four convolutional layers expanding channels from 3 to 512, followed by batch normalization, ReLU, and adaptive average pooling. The results of both backbones are comparable.
- Projector: A three-layer MLP with batch normalization projects the backbone features to a latent space (128 for FashionMNIST, 256 for CIFAR-10).
- Predictor: An encoder-decoder MLP pair:
  - Encoder: Reduces latent dimensions to 64 (FashionMNIST) or 128 (CIFAR-10).
  - Decoder: Reconstructs the latent dimension with sigmoid activation.

TRAINING SETUP:

The model is trained with a batch size of 128, learning rate of 1e-3, and Adam optimizer (weight decay 1e-5). A StepLR halves the learning rate every 10 epochs over 500 epochs. A linear classifier is trained on the frozen representations, and performance is evaluated using Top-1 accuracy for FashionMNIST and Top-2 accuracy for CIFAR-10.

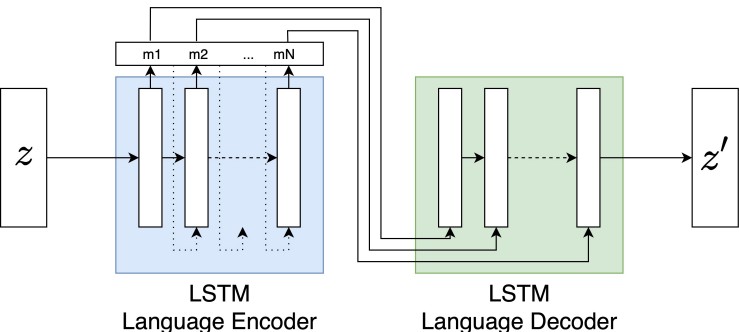

Figure 4: Architecture of the language coder comprising an LSTM-based encoder-decoder structure. The LSTM Encoder generates a sequence of discrete tokens to form the message $w = (m1, m2, ..., mN)$ from a representation $z$ in an autoregressive manner, where each token is sampled from a discrete distribution over the vocabulary $V$. The LSTM Decoder then reconstructs a new representation $z'$ from the sequence of token embeddings, with its final hidden state serving as the output.

COMPARISON MODELS:

We compare SimSiam+VAE against SimSiam and VI-SimSiam, all using the same backbone and projector for fair comparison, with consistent seeding. The predictor encoder of SimSiam+VAE is similar to SimSiam and VI-SimSiam. VI-SimSiam adds a predictor encoder for kappa estimation, while SimSiam+VAE includes a predictor decoder to reconstruct $z'$ from the variable $w$.

# H    EXPERIMENT 2: SIMSIAM NAMING GAME IN EMCOM

DATASETS:

We use the dSprites dataset (Matthey et al., 2017), which contains 2D shape images that vary across different generative factors (e.g., shape, scale, orientation).

MODEL ARCHITECTURE:

- Backbone: is an MLP with three linear layers: the input dimension of 4096 is reduced to 512, then 256, and finally 128. Each layer, except the last, is followed by batch normalization and ReLU activation.
- Projector: A multi-layer perceptron (MLP) projects the feature dimension (128) to a latent dimension of 256 through three fully connected layers. Each layer is followed by batch normalization and ReLU, except for the final layer.
- Predictor: (Language Coder) handles message generation and reconstruction (Fig. 4):
    - Encoder: An LSTM autoregressively generates a sequence of tokens, each sampled from a discrete distribution over a vocabulary of size. At each step, the LSTM produces logits, one-hot vectors, and messages by taking the embedding of the previously generated token as input.
    - Decoder: Another LSTM takes these one-hot token embeddings and processes the sequence to produce a final hidden state that reconstructs the original representation.

TRAINING SETUP:

The vocabulary size $|V|$ is 100, and the message length is 10. The model is trained with a batch size of 256 and a learning rate of 1e-5, using an Adam optimizer. The learning rate is scheduled to decay by half every 10 epochs using a StepLR scheduler. The model is trained for 1000 epochs.

