# OpenReview forum: "SimSiam Naming Game: A Unified Approach for Representation Learning and Emergent Communication"
_ICLR.cc/2025/Conference — ICLR 2025 Conference Withdrawn Submission_

### Official Review · Reviewer_sX1G · 2024-10-28

**Soundness:** 2
**Presentation:** 2
**Contribution:** 2
**Rating:** 3
**Confidence:** 3

**Summary:**

The paper proposes SimSiam+VAE, a model that integrates a Variational Autoencoder (VAE) with the SimSiam framework to incorporate uncertainty in representation learning. This model is extended into the SimSiam Naming Game (SSNG), a framework for emergent communication where agents align their representations and develop a shared language through iterative communication. The approach demonstrates improved performance in both representation learning (on CIFAR10 and FashionMNIST) and emergent communication tasks, outperforming related methods in experiments.

**Strengths:**

The idea of incorporating uncertainty into representation learning with SimSiam is interesting and holds many potential advantages. For example, a probabilistic formulation can be useful in handling stochastic environments. Current SSL methods are not readily able to handle any stochasticity in the data.

**Weaknesses:**

- The paper seems to be exploring two directions: 1) Representation Learning and 2) Emergent Communication. However, it seems that both these directions are not concretely explored or discussed.
- In representations learning, only CIFAR-10 and FashionMNIST are used which are relatively simple and small dataset and its hard to see why SSL would be applicable to such small datasets. Also most standard works use imagenet for comparing SSL algorithms including the VI-Simsiam paper on which this paper builds.
- It would be useful to add a more thorough discussion of emergent communication literature and relevant baselines. Currently, the paper only uses dsprites which is a relatively simple environment. It would be useful to explore more complex environments such as clevr. Moreover, I find that the SimSiam naming game model is much different from the SimSiam VAE model. In the naming game model, it seems that the predictor outputs a distribution over discrete tokens which is different from what SimSiam + VAE does where the predictor outputs the mean and variance of a gaussian. Hence, I do not see how these two model can be considered as a single approach or how SSNG can be considered as an extension of SimSiam + VAE.
- It would be useful to provide a brief description of the TopSim metric in the paper itself.

**Questions:**

- Why use top-2 accuracy instead of top-1 for CIFAR-10? I believe the standard is to report top-1 accuracy.
- In alg. 2 (line 16), what would be the second argument to the KL term?

---

> ### Author Response · Authors · 2024-11-18
>
> Thank you for your detailed feedback.
>
> - We plan to expand our experiments to more complex datasets, including CIFAR-100 and ImageNet100 for representation learning, and CLEVR for emergent communication.
> - Our model seeks to learn p(w,Z|X), where X is the data, Z is the representation, and w is the latent variable from the VAE. The key difference is that in representation learning, w follows a Gaussian distribution (continuous); while in emergent communication, w follows a categorical distribution (discrete), aligning with the nature of messages as sequences of tokens.
> - We will include a brief description of the TopSim metric in the appendix.
> - In our experiments, we use ResNet18 trained from scratch, resulting in lower overall accuracy. As a result, we reported Top-2 accuracy in this version. In future work, we will report Top-1 accuracy with pre-trained models.
> - In Algorithm 2, w follows a categorical distribution with a uniform prior. The KL divergence is computed based on this assumption, comparing the learned categorical distribution with the uniform prior.
>
> Thank you again for your insightful suggestions. We will address these points in our next work.

---

> > ### Comment · Reviewer_sX1G · 2024-11-22
> > **Response to Rebuttal**
> >
> > Thank you for the rebuttal and clarifying my questions. I believe the suggested changes should be incorporated in the current work  as compared to keeping them for future work. In the current state of the paper, given the mentioned weaknesses in my review, I will maintain my score.

---

### Official Review · Reviewer_XN1F · 2024-11-02

**Soundness:** 2
**Presentation:** 3
**Contribution:** 2
**Rating:** 3
**Confidence:** 3

**Summary:**

The paper adopts self-supervised learning to learn emergent communication, enabling agents to develop a shared language. It first enhances self-supervised representation learning VI-SimSiam to propose SimSiam+VAE. This method is then applied to a communication framework called the SimSiam Naming Game (SSNG) to learn emergent communication. The experiments demonstrate that both components achieve comparable or improved performance compared to the baselines.

**Strengths:**

1. It is insightful to link representation learning, such as VI-SimSiam, with emergent communication issues. This approach highlights the connection between these fields and opens new research avenues for training agents to develop shared languages. The idea is both novel and inspiring.
2. The presentation is clear, and the proofs are thorough. The authors have effectively organized the content, making the methodology easy to follow.

**Weaknesses:**

1. With the development of LLMs, communication between agents can now be conducted in natural language, which may offer new pathways beyond traditional methods. However, the authors do not seem to mention this in the paper, which may undermine the significance of the study.
2. The motivation for and effectiveness of VAE are not clearly articulated. In the current version, while it may be natural for the audience to assume that VAE could provide a latent representation for emergent communication, the advantages over VI in the representation learning aspect are not specified. Additionally, there is no experimental analysis comparing the performance of the two methods.
3. Although I do not consider state-of-the-art performance as the sole criterion for evaluating the paper, it is concerning that SSNG achieves comparable performance on dSprites without a detailed explanation.

**Questions:**

1. Is there any specific improvement to VAE that makes it suitable for the proposed method SimSiam+VAE?
2. How would you conclude the core contributions at a theoretical level?
3. Could you provide more explanations for Table 2?
4. In line 81, there should be a space before "O" and a hyphen after it.

---

> ### Author Response · Authors · 2024-11-18
>
> Thank you for your detailed feedback.
>
> - LLMs could offer new directions for agent communication. We will consider this in our next work by replacing the current LSTM with an LLM such as GPT-2.
> - The latent space of the VAE serves as a mapping space from learned representations and is used as shared signs in emergent communication. We will clearly explain this in the paper.
> - Our work provides a framework can be used for both emergent communication (bi-directional interaction game for two agents) and representation learning (with uncertainty based on SimSiam network).
> - Table 2 compares the compositionality of generated messages from our naming game with other communication methods. To measure this, we use TopSim, which calculates the Spearman correlation between pairwise distances in the message space and the corresponding distances in the true generative factors of objects. This metric evaluates how well the message structure aligns with the underlying structure of the data.
>
> Thank you again for pointing out the formatting issue in line 81 and your insightful suggestions. We will address these points in our next work.

---

> > ### Comment · Reviewer_XN1F · 2024-11-26
> >
> > Thank you for your response. Unfortunately, I don't believe the authors have addressed my concerns. There is no explanation regarding Q1 that elaborates on the specific improvements of the VAE presented in this paper. Additionally, for Q2, the authors did not provide a summary of their contributions at a **theoretical** level. Considering the weaknesses mentioned above, as well as comments from other reviewers, I have decided to lower my score.

---

### Official Review · Reviewer_9R6r · 2024-11-03

**Soundness:** 2
**Presentation:** 3
**Contribution:** 3
**Rating:** 6
**Confidence:** 3

**Summary:**

The authors introduce a model for a contrastive self-supervised learning called SimSiam+VAE that builds on previous work, SimSiam and VI-SimSiam. In common with SimSiam approaches, the goal is to learn a representation where similar datapoints are close together (by learning from positive examples alone).

The authors extend previous approaches by integrating a Variational Autoencoder (VAE) into the predictor, enabling the model to better capture uncertainty. Furthermore, by including a VAE, and its KL divergence term, the approach may aid representation learning by regularising the learned representation, which should help prevent collapse and encourage a better spread of representations across the latent space. Building on the model, the authors develop a framework for emergent communication called the SimSiam Naming Game (SSNG), where agents iteratively align their internal representations and develop a shared language

They show results where SimSiam+VAE outperforms SimSiam and VI-SimSiam in experiments on representation learning and emergent communication tasks.

**Strengths:**

Originality and significance: integrating a VAE into SimSiam is novel and builds on previous work expressing SimSiam and similar methods from a Bayesian and variational perspectives. It seems like a well motivated direction: Firstly, the goal of estimating uncertainty with representations is a useful one when thinking about actual applications of the learned representations. Secondly, the particular learning regime in SimSiam – learning to predict what the representation of a similar data point might be – seems less well-posed without explicitly casting that prediction in terms of uncertainty. Finally, the application of this method in an emergent communication framework is an interesting direction and a less explored application.

Quality and clarity: the paper is well written and presented, with a nice flow from motivation through methodology to experiments and places the contributions into the context of prior and related works. Diagrams and schematics are clear and help the reader visualise and understand what's being discussed. And there’s a good level of detail, with all the important elements described for understanding and reproducibility as well as concise supporting code.

**Weaknesses:**

1) Focus on limited and less challenging datasets: given the motivation and promise of the approach, it’s surprising that the experiments are limited to simpler datasets/tasks like FashionMNIST and CIFAR-10. The original SimSiam work was tested on ImageNet so this would be a great candidate for expanding on the experimental results. If the approach was found to struggle on such data & tasks, it would be useful to discuss and explore this. Similarly, comparing the results to other non-SimSiam representation learning approaches on the same datasets would help put the work into a wider context.
2) Rationale: the paper introduces a VAE to model representation uncertainty and improve the learned representations. It could provide a clearer rationale for why a VAE as opposed to other probabilistic models is particularly suited for this in SimSiam. Similarly, the work discusses uncertainty but doesn’t fully explore how this aspect could be applied in downstream tasks where uncertainty estimation would be beneficial, such as anomaly detection or active learning. The work also does not explore the actual performance of the model's uncertainty outputs.
2) Lack of ablations and variations that explore if and how the model benefits from the VAE: some ablation or other experimental variants that help explore and dig into the workings of the model would add to the understanding of the model (also see questions).

**Questions:**

Representation learning results: how does the model perform re representation learning on more challenging datasets / classification tasks like ImageNet. How does the model perform compared to non-SimSiam approaches?

Uncertainty estimation: Are the uncertainty estimates learned from the model accurate and/or reflect something useful or interesting (e.g. relating to epistemic or aleatoric uncertainty)?

How does the model work: given that the stop gradient helps prevent collapse in SimSiam, and the authors speculate that use of a VAE will also help with this, what is the effect of removing the stop gradient in SimSiam+VAE? What is the effect of modifying the weightings on the loss terms, and what can this tell us about how adding a VAE is benefiting the approach. Why did the authors choose different latent distributions in for the representation learning versus emergent communication experiments? Are there other latent distributions that would make sense to use? Other priors that would make sense for the KL divergence term?

---

> ### Author Response · Authors · 2024-11-18
>
> Thank you for your detailed feedback.
>
> - We acknowledge the importance of testing on more challenging datasets. Due to resource limitations, running on ImageNet is currently not feasible, we plan to conduct experiments on CIFAR-100 and ImageNet100. Additionally, we will expand our experiments to include comparisons with non-SimSiam methods to provide a broader context for our results.
> - Similar to beta-VAE, the weight in KL divergence is for disentanglement. We will show the effect of the weight in KL divergence..
> - In our current implementation, the stop-gradient mechanism remains essential for preventing collapse, even with the VAE integrated into SimSiam.
> - For emergent communication, we use a categorical distribution for the latent variable because the generated messages in our model are a series of tokens. In future work, we plan to use Gaussian distributions for the latent space as shared signs of continuous representations (e.g., audio).
>
> Thank you again for your insightful suggestions. We will address these points in our next work.

---

> > ### Comment · Reviewer_9R6r · 2024-11-22
> >
> > Thank you for the follow-up. One brief point regarding the weight in the KL divergence term: another thing that it controls (besides affecting disentanglement) is the strength of the information bottleneck in the representation This may be useful for interpreting any results from experimenting with the weight.

---

### Official Review · Reviewer_tAZe · 2024-11-04

**Soundness:** 2
**Presentation:** 1
**Contribution:** 2
**Rating:** 3
**Confidence:** 3

**Summary:**

SimSiam is a simple SSL objective for images. VI-SimSian (Nakamura et al, 2023) is an extension that replaces the single predictor network with predictor of mean and variance to account for uncertainty. This paper proposes SimSiam+VAE, which extends VI-SimSiam by sampling from a gaussian with the predicted mean and variance. On FashionMNIST and CIFAR10, they show slightly improved performance over SimSiam and SimSiam-VI


The authors also propose an emergent communication setup SimSiam Naming Game. This uses a discrete latent variable and is trained using the Gumbel-Softmax Straight-Through (which maybe doesn't make it a VAE?). They show comparable performance to a regular emergent communication setup on a toy task, dsprites.

**Strengths:**

Integrating discretization and other inductive biases in SSL is an interesting research direction and the authors somewhat approach this.

**Weaknesses:**

This paper never really explains any motivation for adding a VAE into SimSiam and conducts experiments on toy tasks that are not sufficient to empirically demonstrate performance. Furthermore, the paper fails to cite very related work while being blatant in unrelated self-citation, in a way that arguably de-anonymizes the authors.

The only two SSL datasets are CIFAR10 and FashionMNIST and the only baselines are SimSiam, and VI-SimSiam. Work in image SSL should *at minimum* demonstrate results on ImageNet, as done in VI-SimSiam.

Emergent communication results are on little-used dSprites dataset with LSTMs for encoding/decoding. This is very dated and the authors don't even report communication success, the most basic metric for this task. They only show topographic similarity of the message space and find equivalent results to the most simple baseline.

The connection of using SSL methods for emergent communication methods (and vice-versa) was notably done by [Dessi et al, 2021](https://arxiv.org/abs/2106.04258) and feels like a key citation left out. Not as important, but related, is [Lo et al, 2024](https://arxiv.org/abs/2307.01403) that showed an equivalence between contrastive SSL and learning emergent communication in situated MARL. On top of Jang et al (2017) Gumbel-Softmax, you should also cite Concrete Distribution [(Madison et al, 2017)](https://arxiv.org/abs/1611.00712) as it was simultaneous discovery.

(Very probable) self-citation is blatant in the related work section on Emergent Communication. Barely related work on MHNG is cited and it is never explained how it relates to the paper at hand e.g. Citing Inukai et al on recursive multi-agent systems using MHNG. This isn't a huge issue but does look bad.

**Questions:**

Can you run on ImageNet and compare to known SimSiam numbers?
What benefits of the VAE can you *quantitatively* show?

---

> ### Author Response · Authors · 2024-11-18
>
> Thank you for your detailed feedback.
>
> - Motivation for VAE in SimSiam: The VAE introduces uncertainty modeling, provides a generative and probabilistic framework, and facilitates latent space mapping with disentangled representations. In this work, the latent variables act as shared signs in emergent communication. We will mention this in our work.
> - About the experiments: For representation learning, while we lack the resources for ImageNet, we will explore CIFAR-100 and ImageNet100 to further validate performance. For emergent communication, we plan to expand experiments to 3DShapes and CLEVR datasets and explore alternative language model architectures. We will include the quantitative and qualitative of the uncertainty.
> - About citations: We will review your suggestions and include those relevant works. Regarding MHNG, we cited these works because the SimSiam Naming Game closely aligns conceptually and serves as potential future research. We apologize if this caused any misunderstanding.
>
> Thank you again for your insightful suggestions. We will address these points in our next work.

---

### Note · Authors · 2024-11-26

I have read and agree with the venue's withdrawal policy on behalf of myself and my co-authors.